# Solid Lipid Curcumin Particles Protect Medium Spiny Neuronal Morphology, and Reduce Learning and Memory Deficits in the YAC128 Mouse Model of Huntington’s Disease

**DOI:** 10.3390/ijms21249542

**Published:** 2020-12-15

**Authors:** Abeer Gharaibeh, Panchanan Maiti, Rebecca Culver, Shiela Heileman, Bhairavi Srinageshwar, Darren Story, Kristin Spelde, Leela Paladugu, Nikolas Munro, Nathan Muhn, Nivya Kolli, Julien Rossignol, Gary L. Dunbar

**Affiliations:** 1Field Neurosciences Institute Laboratory for Restorative Neurology, Central Michigan University, Mt. Pleasant, MI 48859, USA; maiti1p@cmich.edu (P.M.); me@rebeccaculver.com (R.C.); heile1sj@cmich.edu (S.H.); srina1b@cmich.edu (B.S.); story1dt@cmich.edu (D.S.); speldekr@msu.edu (K.S.); palad1l@cmich.edu (L.P.); munro1nm@cmich.edu (N.M.); muhn1l@cmich.edu (N.M.); nivyareddy91@gmail.com (N.K.); rossi1j@cmich.edu (J.R.); 2Program in Neuroscience, Central Michigan University, Mt. Pleasant, MI 48859, USA; 3Insight Research Center, Insight Institute of Neurosurgery & Neuroscience, Flint, MI 48507, USA; 4Department of Psychology, Central Michigan University, Mt. Pleasant, MI 48859, USA; 5Field Neurosciences Institute, Ascension St. Mary’s, Saginaw, MI 48604, USA; 6College of Health and Human Services, Saginaw Valley State University, Saginaw, MI 48710, USA; 7College of Medicine, Central Michigan University, Mount Pleasant, MI 48859, USA

**Keywords:** Huntington’s disease, curcumin, lipids, YAC128, MSNs

## Abstract

Huntington’s disease (HD) is a genetic neurodegenerative disorder characterized by motor, cognitive, and psychiatric symptoms, accompanied by massive neuronal degeneration in the striatum. In this study, we utilized solid lipid curcumin particles (SLCPs) and solid lipid particles (SLPs) to test their efficacy in reducing deficits in YAC128 HD mice. Eleven-month-old YAC128 male and female mice were treated orally with SLCPs (100 mg/kg) or equivalent volumes of SLPs or vehicle (phosphate-buffered saline) every other day for eight weeks. Learning and memory performance was assessed using an active-avoidance task on week eight. The mice were euthanized, and their brains were processed using Golgi-Cox staining to study the morphology of medium spiny neurons (MSNs) and Western blots to quantify amounts of DARPP-32, brain-derived neurotrophic factor (BDNF), TrkB, synaptophysin, and PSD-95. We found that both SLCPs and SLPs improved learning and memory in HD mice, as measured by the active avoidance task. We also found that SLCP and SLP treatments preserved MSNs arborization and spinal density and modulated synaptic proteins. Our study shows that SLCPs, as well as the lipid particles, can have therapeutic effects in old YAC128 HD mice in terms of recovering from HD brain pathology and cognitive deficits.

## 1. Introduction

Huntington’s disease (HD) is an autosomal dominant progressive neurodegenerative disorder that causes involuntary movements, psychological changes, and cognitive impairment, including dementia [1]. A mutation in the Huntingtin gene (*HTT*) leads to expansion of the cytosine-adenine-guanine (CAG) repeats, which causes an elongation of the polyglutamine segment in huntingtin protein (HTT), resulting in the mutated form of this protein (mHTT) [2].

Although the exact function of HTT is not fully understood, studies have confirmed that HTT interacts with various proteins in cells, helps mediate the transport of neuronal organelles [3,4,5], synaptic vesicles [6,7], and brain-derived neurotrophic factor (BDNF) [8], and plays a role in endocytosis [9,10]. On the other hand, the mHTT induces toxicity in neurons and causes profound damage to the medium spiny neurons (MSNs) in the striatum in the early stages of the disease, while affecting neurons in the cortex and other brain areas during later stages [1].

Different mechanisms have been described to explain the pathogenesis of HD. The loss of BDNF with changes in its receptor (TrkB) is among the most common explanation for the pathological processes occurring in HD [1]. BDNF is a critical neurotrophic factor for the survival of MSNs in the striatal region of the brain, as well as for the maintenance of the synapses in the cortico-striatal pathway [8]. Glutamate excitotoxicity, alterations in presynaptic and postsynaptic proteins, disruption in vesicular transport, and changes in neurotransmitter levels are among the changes that have been observed in HD brains [1,6]. Importantly, it was found that the wild-type HTT interacts with PSD-95, which is a postsynaptic protein, and this leads to sensitization of NMDA receptors. In HD, PSD-95 is released in higher amounts, leading to imbalance in NMDA receptors and excitotoxicity, which leads to damage in synapses and synaptic transmission [1].

Several studies have been conducted using different rodent models of HD to find an effective treatment. We used YAC128 mice, which have the full-length human *HTT* with 128 CAG repeats [11] and that produce motor and cognitive deficits, as well as age-dependent cortical and striatal atrophy [12], which mimic the changes observed in human HD.

Studies have shown that curcumin, a natural polyphenol that is extracted from the plant *Curcuma longa*, has beneficial effects for reducing symptoms in neurodegenerative diseases [13,14]. Curcumin can cross the blood–brain barrier and is known to have antioxidant properties [15]. It has been shown to reduce motor and cognitive impairments in a 3-nitropropionic acid (3-NP) HD rat model [16] and decreases mHTT aggregates, while preserving DARPP-32 and D1 receptors, in the CAG 140 knock-in mouse model of HD [17]. In the 3-NP HD rat model, the administration of curcumin, encapsulated in solid lipid nanoparticles (C-SLNs), increased the activity of mitochondrial complexes and cytochrome levels, resulting in decreased mitochondrial swelling [18].

Due to its poor solubility and low bioavailability, curcumin has limitations in its therapeutic efficacy [19]. Several research studies are ongoing to produce formulae that could improve the solubility and bioavailability of curcumin to increase its efficacy [20]. Recently, we have been using solid lipid curcumin particles (SLCPs; Vendure Sciences, Novelsville, IN, USA), which consist of long-chain phospholipid bilayer and a long-chain fatty acid solid lipid core, which coats the curcumin molecules. This formula contains approximately 26% curcumin, and these SLCPs were characterized in our lab [20], as well as in other laboratories [21]. Dadhaniya and colleagues [22] showed that using a low dose of SLCPs (80 mg/kg/day) produced a protective effect, with no associated adverse effects with doses up to 720 mg/kg/day orally. Therefore, as a proof-of-principle assessment of the potential efficacy of SLCPs as a therapeutic for HD, we treated YAC128 mice with SLCPs or solid lipid particles (SLPs) for eight weeks with 100 mg/kg, orally, every other day, to assess the efficacy for treating HD-like neuropathological and cognitive deficits. We used the active avoidance task in HD-treated mice for the behavioral assessment and compared the performance of wild-type (WT) and YAC128 mice given only vehicle (phosphate-buffered saline; PBS), SLPs without the curcumin, or SLCPs. Morphological changes in MSNs were analyzed using Golgi-Cox staining. Protein analyses, using Western blots for BDNF, TrkB, DARPP-32 (expressed by MSNs), synaptophysin, and PSD-95, were also performed on the brains of these mice.

## 2. Results

### 2.1. Active Avoidance

To test the differences in learning on the active avoidance task, a non-aversive stimulus (tone) immediately preceded an aversive stimulus (shock) and the ability of the mice to escape the aversive stimulus was measured. A repeated-measures ANOVA indicated a significant between-group effect in the latency to escape the shock over all testing trials (*F*_(5, 48)_ = 7.405, *p* < 0.05). PLSD post-hoc tests revealed an overall significant difference in latency to escape between WT+PBS and HD+PBS mice (*p* = 0.01), indicating that HD mice have learning and memory deficits. However, there was no overall significant difference between WT+PBS and HD mice treated with SLCPs (*p* = 0.48), suggesting that the SLCP treatment prevented learning and memory deficits in HD mice. Interestingly, HD mice treated with SLPs also showed an overall significant decrease in the latency to escape compared to WT+PBS and HD+PBS mice (*p* < 0.05), indicating that SLP treatment also improved memory and learning in HD mice.

One-way ANOVAs for each trial revealed significant differences between groups on the first trial of day two (D2T1) and trials on day four and day five. PLSD post-hoc results showed that HD vehicle-treated mice had significantly longer latencies to escape compared to all other WT vehicle mice and all WT- and HD-treated mice on the first trial of day two (D2T1) of testing (*p* < 0.05). There were no significant differences between WT vehicles, WT treated with either SLPs or SLCPs, and HD mice treated with either SLPs or SLCPs (*p* > 0.05). There was a significant decrease in latency to escape in the HD mice treated with SLPs or SLCPs, compared to HD vehicle controls (*p* < 0.05) on day five. Additionally, HD mice treated with SLPs and WT mice treated with SLCPs showed a significant decrease in the latency to escape, compared to other groups, including WT vehicle mice (*p* < 0.05; Figure 1). Multivariate analyses for gender differences within each group revealed no significant differences (*p* > 0.05); therefore, the data for males and females were combined for all analyses.

### 2.2. Golgi-Cox

#### 2.2.1. Dendritic Arborization

To study the effect of the treatment on the morphology of MSNs, Golgi-Cox staining was done, and the length and number of dendrites were compared between groups. One-way ANOVAs showed that there were significant differences in total length (µm) and number of dendritic arborizations of MSNs (*F*_(5, 144)_ = 4.373 and *F*_(5, 144)_ = 7.026; *p* < 0.05, respectively). PLSD post-hoc tests showed greater total length of dendrites in WT vehicle mice, compared to vehicle-treated HD mice (*p* = 0.001) and SLP-treated mice (*p* = 0.018). However, there were no significant differences between WT-vehicle mice and HD mice treated with SLCPs (*p* = 0.153). Additionally, the dendrites in HD mice treated with SLCPs were longer compared to HD-vehicle controls (*p* = 0.05), suggesting that SLCP treatment preserved/increased the total length of dendrites. For the total number of dendrites, the PLSD post-hoc test showed that the number of dendrites in WT-vehicle group was significantly higher compared to the HD-vehicle group, but no significant differences were observed between WT and HD mice treated with SLCPs. There were also significant increases in the number of dendrites in HD mice treated with SLCP compared to HD-vehicle mice, suggesting that SLCPs treatment increased or preserved the number of dendrites in HD mice, while there were no significant differences between HD-vehicle and HD mice treated with SLPs. An analysis of the primary, secondary, and distal dendrites revealed additional differences between groups. One-way ANOVAs showed that there was an overall significant difference between groups in the number and length of primary, secondary, and distal dendrites. The PLSD post-hoc analyses indicated there was a significant difference in the length of primary dendrites between WT- and HD-vehicle mice (*p* = 0.013), but no significant differences in dendritic length between WT and HD mice treated with SLCPs were observed (*p* = 0.162). However, there was no significant difference between HD-vehicle and HD mice treated with SLCPs (*p* = 0.266). Additionally, there was a significant difference in the number of primary dendrites between WT-vehicle and HD mice treated with either vehicle (*p* = 0.001), SLCPs (*p* = 0.023), or SLPs (*p* = 0.001). For the secondary dendritic number and length, there was a significant difference between WT and HD mice treated with vehicle and those treated with SLPs, but no significant difference between WT mice treated with vehicle and HD mice treated with SLCPs was observed. Additionally, there was a significant difference between HD mice treated with vehicle and HD mice treated with SLCPs. For the number of secondary dendrites, there was a significant difference between the WT and HD vehicle groups but no significant difference between WT and HD mice treated with either SLCPs or SLPs. Additionally, HD mice treated with either SLCPs or SLPs showed significantly more secondary dendrites compared to HD-vehicle mice (*p* = 0.001 and *p* = 0.044, respectively). For distal dendrites, there was a significant difference between WT- and HD-vehicle groups in the number and length of dendrites. However, there was no significant difference between WT-vehicle and HD mice treated with SLCPs in the length and number of distal dendrites (*p* = 0.188 and *p* = 0.925, respectively). Additionally, HD mice treated with SLCPs showed significantly higher numbers of dendrites, but not in the length of distal dendrites, compared to the HD-vehicle group (*p* = 0.007 and *p* = 0.064, respectively). HD mice treated with SLPs also showed no significant difference in either number of distal dendritic spines or length for spines, compared to WT (*p* = 0.222) or the HD-vehicle group (*p* = 0.127; Figure 2).

#### 2.2.2. Density of Dendritic Spines

A one-way ANOVA of the dendritic spine density data revealed an overall between-group difference (*F*_(5, 354)_ = 78.120, *p* < 0.05). PLSD post-hoc tests showed that WT groups have more dendritic spines on MSNs compared to the HD groups. However, HD mice treated with either SLCPs or SLPs had significantly greater dendritic spine density when compared to HD mice treated with vehicle (*p* < 0.05). There was also a significant increase in dendritic spine density in WT mice treated with either SLCPs or SLPs, when compared to WT mice treated with vehicle (*p* < 0.05; Figure 3)

### 2.3. Western Blots

#### 2.3.1. DARPP-32

A one-way ANOVA revealed an overall between-group difference (*F*_(5, 12)_ = 12.43, *p* < 0.05) in the amount of DARPP-32, with PLSD post-hoc tests indicating significant decreases in DARPP-32 levels in the HD groups compared to WT groups (*p* < 0.05), corresponding to a significant reduction of MSNs in HD mice. Additionally, there were no significant differences between HD-vehicle and HD mice treated with either SLCPs or SLPs (*p* > 0.05; Figure 4A,B), indicating that SLCP or SLP treatments did not preserve the number of MSNs in HD mice.

#### 2.3.2. BDNF

A one-way ANOVA for BDNF labeling indicated no overall significant difference between groups (*F*_(5, 12)_ = 2.765, *p* = 0.069; Figure 5A,D). However, a trend toward reduced BDNF levels in vehicle- treated HD mice was observed.

#### 2.3.3. TrkB

One-way ANOVAs revealed an overall between-group difference in truncated TrkB levels (*F*_(5, 12)_ = 3.148, *p* = 0.048) but no significant differences in long TrkB levels (*F*_(5, 12)_ = 0.916, *p* = 0.503). PLSD post-hoc tests of truncated TrkB levels revealed a trend toward decreasing TrkB levels in HD controls compared to WT controls (*p* = 0.056). However, HD mice treated with either SLCPs or SLPs were not significantly different from WT controls (*p* = 0.822 and *p* = 0.938, respectively), indicating the preservation of TrkB levels in HD-treated mice. Additionally, WT mice treated with either SLCP or SLP had significantly higher levels of truncated TrkB than the HD-vehicle group (*p* = 0.010 and *p* = 0.003, respectively), but not significantly different from HD mice treated with SLCPs or SLPs (*p* = 0.084 and *p* = 0.064, respectively; Figure 5B,C,E).

#### 2.3.4. Synaptophysin

One-way ANOVAs revealed no overall significant differences between groups for synaptophysin levels (*F*_(5, 12)_ = 2.537, *p* = 0.087; Figure 6A,D).

#### 2.3.5. PSD-95

A one-way ANOVA revealed the overall between-group difference for PSD-95 levels (bands at 110 kDa; *F*_(5, 12)_ = 4.306, *p* = 0.018). PLSD post-hoc tests indicated no significant difference between WT and HD vehicle groups (*p* = 0.536). However, HD mice treated with SLCPs showed a significant decrease in PSD-95 when compared to HD-vehicle mice (*p* = 0.007) and HD mice treated with SLPs (*p* = 0.021), but not when compared to the WT-vehicle group (*p* = 0.177). A one-way ANOVA showed an overall significant difference between groups for PSD-95 (bands at 80 kDa; *F*_(5, 12)_ = 4.027, *p* = 0.022). PLSD post-hoc test showed no significant differences between WT- and HD- vehicle groups (*p* = 0.108). However, HD mice treated with SLCPs showed a significant decrease in PSD-95 levels compared to HD-vehicle mice (*p* = 0.008), HD mice treated with SLPs (*p* = 0.046), WT mice treated with SLCPs (*p* = 0.037), and WT mice treated with SLPs (*p* = 0.001). Additionally, there was a significant increase in PSD-95 levels in WT mice treated with SLPs when compared to the WT vehicle group (*p* = 0.020; Figure 6B,C,E).

## 3. Discussion

Curcumin is known to have antioxidant properties, and various studies have suggested curcumin as a potential treatment for different neurodegenerative diseases [14,23]. However, the low water solubility, poor bioavailability, and extensive metabolism of curcumin limit the potential health benefits with natural unaltered curcumin [24]. The current study analyzed the effects of a formula that incorporates curcumin into solid lipid particles (SLCPs) as a potential treatment for HD-like deficits in YAC128 mice. Previous studies showed that SLCPs improved the pharmacokinetics of curcumin compared with a generic curcumin extract, suggesting a potential therapeutic effect [21,25]. A comparison between curcumin and SLCPs in our laboratory previously showed that SLCPs provide greater neuroprotective effects [20]. In this study, we analyzed the effects of SLCPs and SLPs (without curcumin) on HD pathology and behavioral deficits in the YAC128 mouse model of HD. Major outcomes of the study include (1) reduced latency to escape in the active-avoidance task in SLCP- and SLP-treated mice, (2) increased dendritic arborization and number of dendritic spines on MSNs in HD mice treated with SLCPs or SLPs, (3) reduced PSD95 levels in HD mice treated with SLCPs compared to the HD control mice, and (4) increasing TrkB levels in the SLCP and SLP, along with modest increases in BDNF levels in SLCP-treated HD mice.

Results from the active-avoidance task showed that HD mice treated with SLCPs had similar latency to escape times when compared to WT mice. These data suggest that the SLCP treatment prevented learning and memory deficits in HD mice, which is consistent with the known neuroprotective properties of curcumin [13]. Additionally, HD mice treated with SLPs without curcumin performed better than vehicle-treated HD mice. This surprising result suggests that SLPs alone confer beneficial effects. This may be due to the lipid contents of the SLPs and SLCPs and the amount of the lipids that were delivered to the brain. Ueda and colleagues [26] showed that dietary lipids can increase memory in a mouse model of aging. Normalizing the lipid composition in the mitochondria can reduce memory deficits in the 3-NP model of HD [27]. Taken together, these studies suggest soluble lipid particles may improve memory in aged YAC 128 mice. Further research is needed to discern how SLPs are exerting their beneficial effects and to what extent this can interact with the effects of curcumin alone.

Our histological results revealed an SLCP-induced preservation of the morphology of the surviving MSNs in the brains of HD mice. This included the number and length of dendrites and dendritic spine densities of MSNs in HD mice treated with SLCP. However, Western blot data showed that there were reductions in DARPP-32 levels in brains of all treated and control HD mice groups compared to WT mice, which suggests a reduction in number of MSNs in HD mice. Even though curcumin is known to have neuroprotective effects and help in the preservation of neurons [20,28], we think that the starting point of treatment is a critical factor that needs to be taken into consideration for optimal neural protection and behavioral outcomes. Earlier treatment, before the onset of symptoms, might help prevent neuronal death, thereby preserving the number of existing MSNs. The rescue of dendritic spines that was found in this study following SLCPs treatment, which is in agreement with previous studies showing the neuroprotective properties effect of curcumin [29,30]. It is also interesting to note that mice treated with SLPs showed a mild preservation of MSN morphology compared to the HD controls. It has been shown that lipids facilitate neurotransmission and have an effect on the interaction of various molecules, including their trafficking and signaling at the synaptic junctions [31], which may help to explain the positive effects of SLPs, alone, on the dendritic arborization and the spinal density observed. Overall, our results showed that SLCPs preserved the morphology of the residual MSNs in the striata of HD brains and suggest that treatment at early stages of the disease may help protect against the loss of larger numbers of MSNs in the striata of HD brains.

Given that synaptic dysfunction occurs in HD, leading to improper neural connectivity [32], we analyzed the levels of presynaptic (synaptophysin) and postsynaptic (PSD-95) proteins to assess the impact of SLCPs and SLPs on neural connectivity. Briefly, synaptophysin is a membrane-associated glycoprotein that is present in the presynaptic vesicles of neurons [33], and PSD-95 is present in the postsynaptic membrane of neurons, and the NMDA receptors are a component of this protein [34]. Although no significant changes were observed for synaptophysin levels in HD-treated mice, a decrease in PSD-95 in SLCP-treated mice was found. This decrease in PSD-95 may be relevant for treating HD, as previous studies have shown that PSD-95 is associated with NMDA receptor-mediated excitotoxicity [35]. Fan and colleagues [36] have shown that there is increased excitotoxicity mediated by PSD-95 in YAC 128 HD mice, which is one hypothesis proposed to explain the pathology of the disease. It has been proposed that the postsynaptic region is a lipid raft-enriched area associated dynamically with PSD-95 [37]. These rafts, including certain lipids, are proposed to have a critical role in synaptic signaling, plasticity, and maintenance by contributing to the accuracy of the spatial and temporal organization of molecules in dendritic spines [37,38]. These results could also support the findings of an increase in dendritic spine density and improvements in learning and memory that were observed after SLCP treatment in this study. Further analyses of different presynaptic and postsynaptic proteins are essential to understanding the changes at the synaptic level following SLP or SLCP treatment, since both treatments seem to induce effects in different pathways.

BDNF is another molecule that is deficient in HD, and its absence can lead to neuronal death. Decreased levels of BDNF in the striatum have been frequently observed in HD [39,40]. Our previous studies have shown that increasing BDNF levels in the HD brain reduces symptoms of the disease, proving that increasing the levels of BDNF is therapeutic in HD [41,42]. In this study, we quantified the levels of BDNF in the entire brain of the treated and untreated HD and WT mice. Our results indicated that there was a trend towards decreasing BDNF levels in HD brains compared to WT brains. Although we did not find a significant change in whole-brain levels of BDNF, the quantification of the truncated TrkB, which is a BDNF receptor, showed that there is an increase in receptor levels following SLCP or SLP treatments compared to the HD controls. TrkB has been proposed to play an important role in HD pathogenesis, and several studies have shown that the modulation of BDNF-TrkB signaling can be significant in HD treatments [43]. Our study suggests that SLCP or SLP treatments may modulate BDNF-TrkB signaling levels, which, in turn, could help in reducing the pathological changes that are seen in HD mice.

## 4. Materials and Methods

### 4.1. Animals and Treatment

All procedures involving animals that were used in this study were approved by the Central Michigan University Institutional Animal Care and Use Committee (protocol # 18-23, on 20 September 2018. Fifty-four (36 males and 18 females) YAC128 (transgenic HD) and WT mice were utilized at 11 months of age, in which six males and three females were randomly assigned to the following subgroups: WT vehicle controls (WT + PBS; *n* = 9), YAC128 vehicle controls (HD + PBS; *n* = 9), WT treated with SLCPs (WT + SLCP; *n* = 9), YAC128 treated with SLCPs (HD + SLCP; *n* = 9), WT treated with SLPs (WT + SLP; *n* = 9), and YAC128 treated with SLPs (HD + SLPs; *n* = 9). Solid lipid curcumin particle (SLCP) is also known commercially as Longvida Optimized Curcumin, which was developed, patented, and licensed by Verdure Sciences (Noblesville, IN, USA). They utilized their Solid Lipid Curcumin Particle (SLCP™) Technology to coat the curcumin under specific conditions with lecithin (phospholipid) and stearic acid. The appropriate dose of SLCPs or SLPs (100 mg/kg) for every mouse was suspended in PBS and administered by oral gavage using polyurethane feeding tubes, 16 ga/38 mm model for viscous compounds (Instech Laboratories, Plymouth Meeting, PA, USA). The gavage treatments were done in the morning, every other day, for 8 weeks. Vehicle controls were treated with PBS (equal volume to those used in the SLCPs dosing). The dose and route of administration were determined and was based on a previous efficacy and safety study [22]. All mice were housed in cages in the vivarium, with 2–4 mice in each polyethylene bin. All mice had a 12-h light/12-h dark cycles and *ad libitum* access to water and food. Examiners were blinded to the group identity of the mice throughout all the testing procedures.

### 4.2. Active Avoidance

The active avoidance task is consisted of a step-through box with two chambers (Hamilton-Kinder LLC; LM1000SP, San Diego, CA, USA)—one dark, and one lit by a house light—positioned above the chamber, with a sliding door separating the chambers. The task was performed for all mice (36 males and 18 females, with 6 males and 3 females in each group). Acclimation was done at the beginning of week 8 of treatments (Day 1 of the task). The mice were placed in the lighted chamber and allowed to explore both chambers for a total of 10 min, with the stipulation that, when the mouse entered the dark chamber, the sliding door would close, and the mouse remained in the chamber for the rest of the 10 min. Training began on the following day, and the mice were placed in the lit chamber and allowed to explore both chambers, with the sliding door open. A tone occurred 5 s after placing the mouse in the lit chamber and continued for 5 s. Following the tone, the mouse was subjected to a 0.5 mA shock, lasting for 2 s. The mouse remained in the chamber for 10 s following the shock, and then it was placed back into its container. If the mouse moved to the dark chamber before the tone ended, the shock could be avoided. This protocol was conducted for 3 trials on day 2 and one trial each on each of the next three days. The latency to escape into the dark chamber was the primary dependent variable.

### 4.3. Golgi-Cox Staining

Three mice (two males and one female) from each group were randomly selected and were euthanized by cervical dislocation, and their brains were extracted and immersed in fresh Golgi-Cox solution for 48 h—after which, the brains were transferred into another fresh Golgi-Cox solution for 14 days. Golgi-Cox staining was performed following a previously published protocol [44]. Golgi solution was made by adding 5 volumes of 5% potassium dichromate in double-distilled water, 5 volumes of 5% mercuric chloride dissolved in double-distilled water, and 4 volume parts of 5% potassium chromate diluted in 10 volume parts of double-distilled water, all mixed with continuous stirring. After two weeks, the Golgi-Cox solution was removed, and a 30% sucrose solution was added, and the brains were kept at 4 °C for 24 h. Then, the brains were sectioned using a vibratome (Ted Pella Inc., Redding, CA, USA) in 120-μm-thick coronal sections that were mounted onto 0.5% gelatin-coated microscope slides. The slides were kept for 48 h on warm water bath; then, they were washed twice for two minutes with double-distilled water and then immersed in 75% ammonia solution for 10 min in the dark. Then, the sections were washed by double-distilled water six times, 5 min each. After that, the sections were immersed in 1% sodium thiosulfate for 10 min, and then, they were washed 6 times, 5 min each, with double-distilled water. The slides were gradually dehydrated using a series of 50%, 75%, and 95% ethanol for 4 min each and then 2 times for 2 min each in 100% ethanol. After that, the sections were immersed in xylene 2 times, for 2 min each, and then in fresh xylene for 1 h. Then, the sections were covered using DePeX (BDH, Batavia, IL, USA) and dried for 48 h at room temperature. Twenty-five neurons from every group in the dorsal striatum were captured randomly using a 20× objective light microscope (Olympus BX51 microscope, Olympus Corporation, Tokyo, Japan). The number and length of dendrites of these neurons were measured using Image-J software (NIH, Bethesda, MD, USA). Fifty distal dendrites on MSNs in the dorsal striatum from each group were captured randomly using a 100× objective. The number of dendritic spines and the length of their corresponding dendrites also were measured using Image-J software (NIH, Bethesda, MD, USA). Examiners were blinded to the group identity of the mice throughout all the imaging and assessment.

### 4.4. Western Blot (WB)

For protein analyses, three mice (two males and one female) from each group were randomly selected and sacrificed by cervical dislocation, and brains were extracted and flash-frozen. Whole brains were lysed in cold radioimmunoprecipitation assay (RIPA) buffer (10-mM Tris-Cl (pH 8.0), 1-mM EDTA, 0.5-mM EGTA, 0.1% SDS, 140-mM NaCl, 0.1% sodium deoxycholate, and 1% Triton X-100, with protease inhibitors (Sigma, St. Louis, MO, USA)). The homogenate was centrifuged at 20,000× *g* at 4 °C for 30 min. The supernatant was taken and aliquoted in PCR tubes and stored at −80 °C, until use. Protein concentrations in each sample were determined using the Pierce Bicinchoninic acid assay (BCA) protein assay (Thermo Scientific, Rockford, IL, USA). Samples were mixed with equal amounts of 2X SDS-sample buffer (125-mM Tris-HCl, pH 6.8, 4% sodium dodecyl sulfate, 20% glycerol, 10% 2-mercaptoethanol, and 0.2% bromophenol blue) and boiled for 5 min. For assessment, an equal amount of protein from each sample was loaded and separated on Tris-glycine gels (4–20%; Invitrogen, Carlsbad, CA, USA). The gel was run at 100 V with a running buffer (25-mM Tris-Base, 192-mM glycine, and 0.1% SDS). The protein from the gel was transferred overnight to a Polyvinylidene difluoride (PVDF) membrane (Millipore, Billerica, MA, USA) in an ice-cold buffer containing 25-mM Tris–Base, 192-mM glycine, and 10% methanol. Following transfer, the blots were rinsed three times in Tris-buffered saline with Tween 20 (TBST), and the membranes were blocked with 5% fat-free milk in TBST for 1 h. Then, the blots were incubated with primary antibodies: rabbit anti-DARPP-32 (1:1000, Abcam, Cambridge, UK) to label MSNs, rabbit anti-BDNF (1:1000; Sigma, St. Louis, MO, USA), rabbit anti-TrkB (1:1000; Cell Signaling Technology, Danvers, MA, USA), mouse anti-PSD-95 (1:300; Santa Cruz Biotechnology, Dallas, TX, USA), rabbit anti-synaptophysin (1:1000; Cell Signaling Technology, Danvers, MA, USA), and rabbit anti- β-tubulin (1:1000; Cell Signaling Technology, Danvers, MA, USA) in 5% fat-free milk powder, dissolved in TBST, and kept overnight at 4 °C. Membranes were then rinsed three times with TBST and incubated with the respective horseradish peroxidase (HRP)-conjugated secondary antibodies (goat anti-rabbit immunoglobulin G (IgG, 1:10,000) or goat anti-mouse IgG (1:5000)) in 1.5% fat free milk powder in TBST for 1 h. The membranes were then washed three times with TBST, and the blots were developed with ImmobilonTM Western Chemiluminescent HRP-substrate (Millipore, Billerica, MA, USA) and scanned. The optical density (OD) of each lane of the blot was measured using Image-J software (NIH, Bethesda, MD, USA) and normalized with the OD of β-tubulin.

### 4.5. Statistical Analyses

All statistical analyses were performed using SPSS v24 (IBM, Armonk, NY, USA). The latency to escape in the active avoidance task was analyzed using repeated measures analysis of variance (ANOVA). Data from each trial was analyzed using one-way ANOVAs. Multivariate analyses were used to test if there were within-group gender differences on data for the active-avoidance task. Number of dendritic spines, length and number of distal dendrites, and Western blot data were analyzed using one-way ANOVA. Fisher’s protected least significant differences (PLSD) post-hoc tests were performed when the omnibus F-values were significant to provide between-group comparisons. The alpha level was set at *p* < 0.05 for all analyses.

## 5. Conclusions

Our study showed that SLCP treatment was able to preserve the existing MSN morphology, modulate synaptic proteins, and improve learning and memory in the YAC128 mouse model of HD. Additionally, SLP treatment, alone, had a marked effect on improving learning and memory in the YAC128 mouse model of HD. Histological and protein analyses suggest that, despite both SLCP and SLP treatments showing beneficial effects, the neuronal pathways responsible for their efficacy may differ. Regardless, the results of the present study suggest that SLCPs and SLPs might offer significant potential as therapeutic agents in maintaining function and delaying the onset of symptoms in HD. Further exploration into the pathways responsible for these effects, as well as the timing of treatment administration, warrant further testing to fully assess the clinical potential of these treatments.

## Figures and Tables

**Figure 1 ijms-21-09542-f001:**
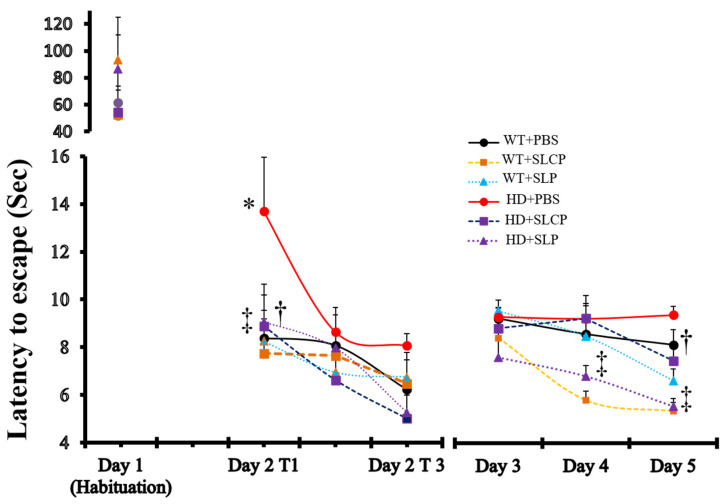
Effects of solid lipid curcumin particles (SLCPs) on the cognitive functions in YAC128 Huntington’s disease (HD) mice. Repeated-measures ANOVA of active-avoidance data showed a significant between-group effect in the latency to escape the shock (*p* < 0.05). Protected least significant differences (PLSD) post-hoc test showed an overall significant difference in latency to escape between wild-type+phosphate-buffered saline (WT+PBS) and HD+PBS mice (*p* = 0.01). There was no significant difference between WT+PBS and HD treated with SLCPs (*p* = 0.48). Additionally, HD mice treated with solid lipid particles (SLPs) showed significant decreases in latency to escape compared to WT+PBS ((*p* = 0.02) and HD+PBS mice (*p* = 0.00). * *p* < 0.05, HD+PBS mice compared to WT groups. † *p* < 0.05, HD mice treated with SLCPs compared to HD+PBS mice. ‡ *p* < 0.05, HD treated with SLPs mice compared to HD+PBS mice.

**Figure 2 ijms-21-09542-f002:**
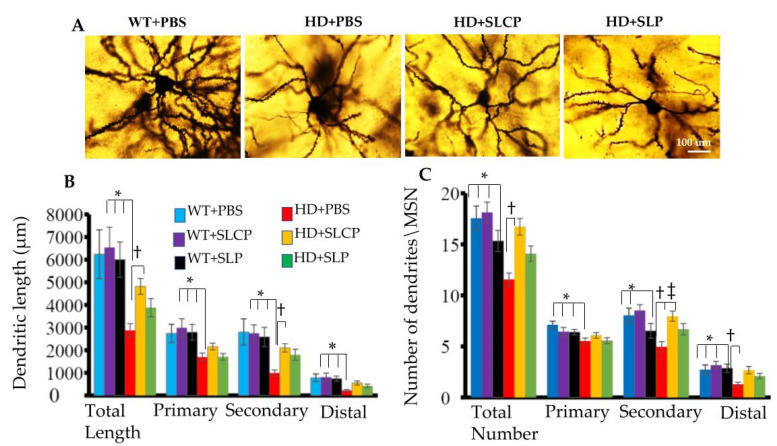
Morphology of medium spiny neurons (MSNs). (**A**) Golgi-cox staining of MSNs from all groups showed differences in the number and length of dendrites between different groups. Images in this figure were taken using a light microscope at 40× objective. (**B**) Total dendritic length, primary, secondary, and distal dendrite lengths. (**C**) Total number of dendrites, primary, secondary, and distal dendrites for every MSN from all groups. (**B**,**C**) There was a significant difference in the total length of dendrites between WT and HD vehicles (*p* = 0.001). However, HD mice treated with SLCPs were not significantly different compared to WT controls (*p* = 0.153). Additionally, HD mice treated with SLCPs were marginally significant different from HD controls (*p* = 0.05). For the total number of dendrites, there was a significant difference between WT- and HD-vehicle groups (*p* < 0.05), but no significant differences between WT and HD mice treated with SLCPs. * *p* < 0.05, HD+PBS compared to WT+PBS. † *p* < 0.05, HD+SLCPs compared to HD+PBS. ‡ *p* < 0.05, HD+SLP compared to HD+PBS.

**Figure 3 ijms-21-09542-f003:**
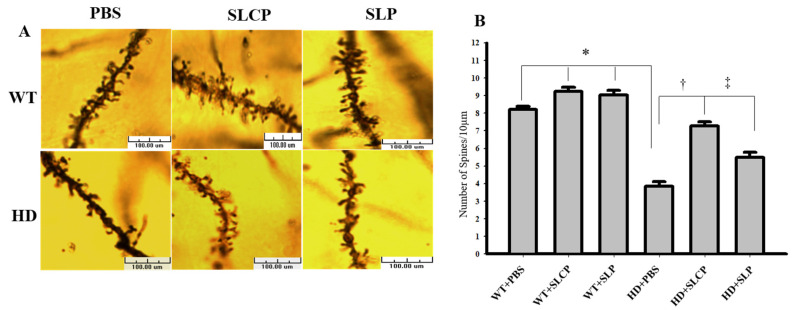
Golgi-Cox staining; dendritic spines of MSNs. (**A**) Images of dendritic spines at 100× objective using a light microscope showing the density of dendritic spines in MSN dendrites from different groups of mice. (**B**) PLSD post-hoc tests showed that there was a significant difference between WT and HD vehicle controls. However, HD mice treated with either SLCPs or SLPs had significantly more dendritic spines compared to the HD-vehicle group (*p* < 0.05). * *p* < 0.05, HD+PBS compared to WT+PBS. † *p* < 0.05, HD+SLCP compared to HD+PBS. ‡ *p* < 0.05, HD+SLP compared to HD+PBS.

**Figure 4 ijms-21-09542-f004:**
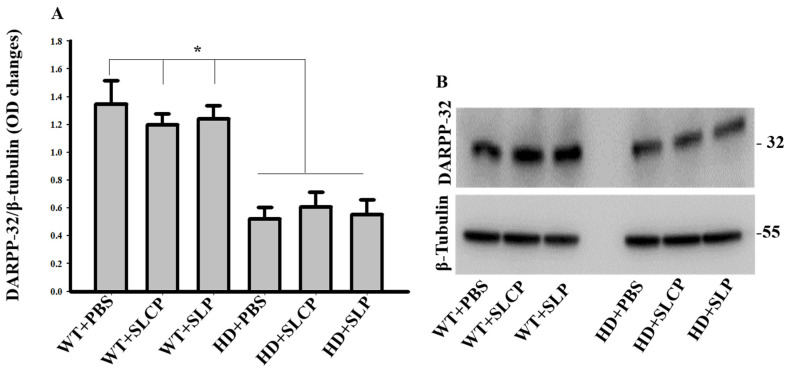
Western blots of DARPP-32. (**A**,**B**) There was a significant elevation in DARPP-32 levels in WT groups compared to HD groups, but there were no significant differences between HD+PBS, HD+SLCPs, and HD+SLPs groups. * *p* < 0.05 between all HD groups compared to all WT groups.

**Figure 5 ijms-21-09542-f005:**
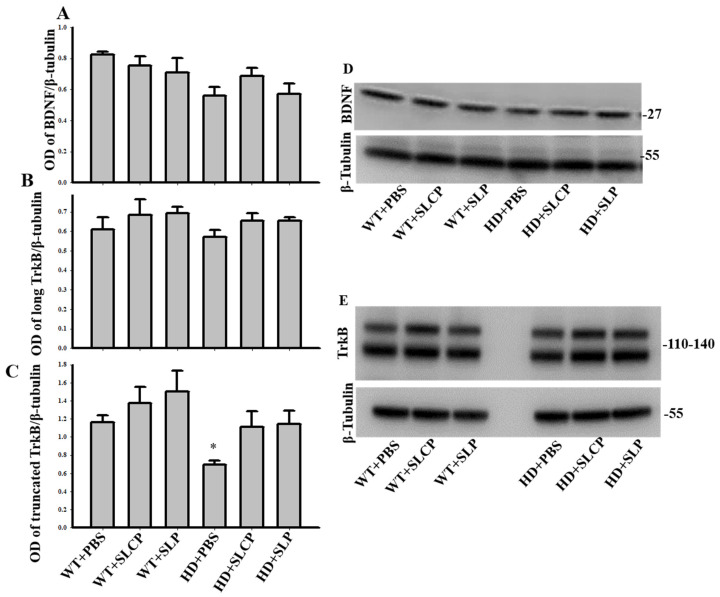
Western blots of brain-derived neurotrophic factor (BDNF) and TrkB. (**A**,**D**) A one-way ANOVA of BDNF data revealed no overall between-group difference (*p* = 0.069). (**B**,**C**,**E**) There was no overall between-group difference in long TrkB (*p* = 0.503), but treatments of either SLCPs or SLPs prevented significant declines in levels of truncated TrkB in HD mice. * *p* < 0.05, HD+PBS compared to WT+SLCP and WT+SLP.

**Figure 6 ijms-21-09542-f006:**
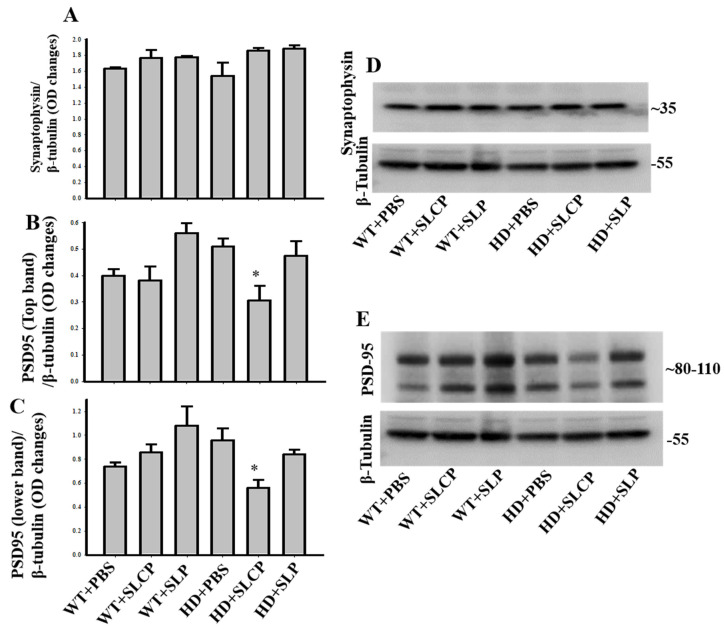
Western blots of synaptophysin and PSD-95. (**A**,**D**) A one-way ANOVA of synaptophysin levels showed no overall significant difference between groups (*p* = 0.089). (**B**,**C**,**E**) Upper and lower bands of PSD-95 revealed no significant differences between WT- and HD-vehicle groups (*p* > 0.05). However, HD+SLCPs mice had significant decreases in PSD-95 levels compared to HD+PBS and HD+SLP (*p* < 0.05), but were not different compared to WT+PBS. * *p* < 0.05, HD+SLCP compared to HD+PBS.

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
