# Peer review of "Solid Lipid Curcumin Particles Protect Medium Spiny Neuronal Morphology, and Reduce Learning and Memory Deficits in the YAC128 Mouse Model of Huntington’s Disease"

_ijms, 2020, doi:10.3390/ijms21249542_

Round 1
Reviewer 1 Report
Ref: ijms-1027834
Title: Solid lipid curcumin particles protect medium spiny neuronal morphology, and reduce learning and memory deficits in the YAC128 mouse model of Huntington’s disease.
Recommendation: MAJOR REVIEW
- For each test, please provide information on how many animals of which sex were used.
- How SLCPs works according to the sex of the animals?
- The Authors should add to the methodology description how they performed the SLCP.
- Fifty-four animals have been used. How many animals of which sex were used?
- In the introduction, the Authors should describe the role of DARPP-32, BDNF, TrkB, PSD-95, synaptophysin in the context of HD pathophysiology.
- Figures have very poor resolution.
Author Response
We would like to thank the reviewer for their valuable comments and critiques that helped improving the manuscript. We have revised the manuscript taking in consideration all the points that were raised by the reviewer.
- For each test, please provide information on how many animals of which sex were used.
- Even though sex of the mice was not a significant factor, we agree with the reviewer that this should be explained in the text. As such, we included the distribution of males and females used in the study and included this in the Methods section, Lines 111-112, 125, and 152
- How SLCPs works according to the sex of the animals?
- Again, we agree with the reviewer that this should be explained in the text. As such, we added the following statement on lines 209-211: “Multivariate analyses for gender differences within each group revealed no significant differences, therefore the data for males and females were combined for all analyses”.
3. The Authors should add to the methodology description how they performed the SLCP.
- Solid lipid curcumin particle (SLCP) is also known commercially as Longvida Optimized Curcumin, which was developed, patented, and licensed by Verdure Sciences ( Noblesville, IN). They utilized their Solid Lipid Curcumin Particle (SLCP™) Technology to coat the curcumin under specific conditions with lecithin (phospholipid) and stearic acid. This was added to the methodology section, Lines 95-99. Also, we added that polyurethane feeding tubes, 16ga/38mm model for viscous compounds (Instech Laboratories) were used for gavage. This was added to the methods section, Lines 100-102.
- Fifty-four animals have been used. How many animals of which sex were used?
- 36 males and 18 females were used; six males and three females in each group. This was added to the methods section, line 93-94.
- In the introduction, the Authors should describe the role of DARPP-32, BDNF, TrkB, PSD-95, synaptophysin in the context of HD pathophysiology.
- A paragraph describing the role of MSNs (expressing DARPP-32), BDNF, TrkB, PSD95 and synaptophysin was added to the Introduction section, Lines 45-47. Also, a line (85) was added to specify what was done in the introduction and adding that MSNs express DARPP-32.
6. Figures have very poor resolution.
- Figures with better resolution were added to the revised manuscript. Additional separate copies of the figures are also attached separately.
Reviewer 2 Report
Huntington's disease is a devastating neurodegenerative genetic disease that does not yet have any treatments with significant efficacy. The authors here study treatment of HD with solid lipid curcumin particles (SLCP) in the YAC128 HD mouse model in order to improve bioavailability of curcumin in the brain, as this compound has been previously shown to improve neurodegenerative symptoms in other models. The authors study the effects of SLCP on a behavioral phenotype in their HD mouse model, along with examining effects on changes in neuronal structure and levels of relevant proteins. The results show that while the SLCP treatment has positive effects on the HD phenotype, neuronal structures, and protein levels in the YAC128 mouse model, solid lipid particles without curcumin also show positive effects, indicating the lipid molecule itself may have a therapeutic effect. These results provide the basis to pursue additional studies on the potential therapeutic effects of these molecules in HD.
The experiments are conducted with appropriate controls, with clearly explained methodology and results with appropriate description and data analysis. the text of the paper is written with clarity that make it easy to follow the purpose of the study, the line of experimentation, and a thorough discussion of the results, conclusions, and future directions. I would recommend publishing this paper in its current form and there are not any significant revisions that I would recommend.
Author Response
We thank the reviewer for these comments.
Round 2
Reviewer 1 Report
Recommendation: Accept for publication
After taking into account comments/suggestions, in my opinion the manuscript is ready for publication.